# Sources of stress and coping mechanisms: Experiences of maternal health care providers in Western Kenya

Monica Getahun[1][⊙]*, Edwina N. Oboke[2][‡], Beryl A. Ogolla[2][‡], Joyceline Kinyua[3], Linnet Ongeri[3], Mona Sterling[1], Iscar Oluoch[4], Audrey Lyndon[5], Patience A. Afulani[1,6][⊙]

**1** Institute for Global Health Sciences, University of California, San Francisco, San Francisco, California, United States of America, **2** Global Programs for Research and Training, Nairobi, Kenya, **3** Kenya Medical Research Institute, Nairobi, Kenya, **4** Migori County Government, Migori, Kenya, **5** NYU Rory Meyers College of Nursing, New York, NY, United States of America, **6** Department of Epidemiology & Biostatistics, University of California, San Francisco, San Francisco, California, United States of America

⊙ These authors contributed equally to this work.
‡ ENO and BAO also contributed equally to this work.
* Monica.Getahun@ucsf.edu

**Data Availability Statement:** Due to confidentiality agreements, supporting data can only be made available to bona fide researchers. Details of the data and how to request access are available at

## Abstract

The dynamic and complex nature of care provision predisposes healthcare workers to stress, including physical, emotional, or psychological fatigue due to individual, interpersonal, or organizational factors. We conducted a convergent mixed-methods study with maternity providers to understand their sources of stress and coping mechanisms they adopt. Data were collected in Migori County in western Kenya utilizing quantitative surveys with n = 101 maternity providers and in-depth interviews with a subset of n = 31 providers. We conducted descriptive analyses for the quantitative data. For qualitative data, we conducted thematic analysis, where codes were deductively developed from interview guides, iteratively refined based on emergent data, and applied by a team of five researchers using Dedoose software. Code queries were then analysed to identify themes and organized using the socioecological (SE) framework to present findings at the individual, interpersonal, and organizational levels. Providers reported stress due to high workloads (61%); lack of supplies (37%), poor salary (32%), attitudes of colleagues and superiors (25%), attitudes of patients (21%), and adverse outcomes (16%). Themes from the qualitative analysis mirrored the quantitative analysis with more detailed information on the factors contributing to each and how these sources of stress affect providers and patient outcomes. Coping mechanisms adopted by providers are captured under three themes: addressing stress by oneself, reaching out to others, and seeking help from a higher power. Findings underscore the need to address organizational, interpersonal, and individual level stressors. Strategies are needed to support staff retention, provide adequate resources and incentives for providers, and ultimately improve patient outcomes. Interventions should support and leverage the positive coping mechanisms identified.

https://www.icpsr.umich.edu/web/pages/ICPSR/
index.html.

**Funding:** This study is funded by a Eunice Kennedy
Shriver National Institute of Child Health and
Human development K99/R00 grant to PA
[K99HD093798/ R00HD093798]. The funders had
no role in the study design, data collection and
analysis, decision to publish, or preparation of the
manuscript.

**Competing interests:** The authors have declared
that no competing interests exist.

## Introduction

The dynamic and complex nature of healthcare provision predisposes health care workers to numerous stressors, resulting in behavioral, psychological and physiological stress responses [1–3]. These stressors, some of which providers may or may not have control over, have been well-described, and include high work load, unfavourable work environment and work conditions, witnessing negative outcomes among patients, and low rewards and incentives [4–8]. Continuous exposure to stressors without adequate coping mechanisms leads to burnout, which manifests as feelings of pessimism, cynicism, helplessness, detachment from one's job, and reduced professional efficacy [3]. Burnout affects job performance as well as the psychological and physical health of providers [3, 9–11].

While high stress and burnout among healthcare workers is a recognized global crises [12], providers in sub-Saharan Africa (SSA) are doubly burdened due to the additional stressors of working in resource-limited settings [13]. In addition to similar on-the-job stressors as providers in resource-rich settings, they also manage stressors of an overwhelming work load from extreme staff shortages, high patient volumes, inability to provide best practice due to lack of drugs, supplies, and equipment, management of complications beyond their competency, financial strain from poor remuneration, poor working conditions with insufficient basic resources, including scarcity of water and sanitation, and disrespectful and violent behaviors of patients [14–16]. Providers in SSA are a persistently high-risk group due to continuous exposure to these difficult situations; as many as 9 out of 10 SSA healthcare workers experience some degree of burnout in some settings [17–20].

Our study focuses specifically on maternal health providers, who by the nature of their work, face especially stressful working conditions. In addition to the stressors described, providers often manage double the patient load, coupled with feelings of inadequacy in the face of high maternal and new-born mortality [15]. Studies in Kenya and elsewhere have shown very high levels of burnout among maternity providers which impacts care quality [19–21]. The adverse impact of stress and burnout on patient experience and other patient outcomes have also been well documented [22–27]. Addressing stress and burnout in maternal health is thus an opportunity to promote person-centred maternal care, and improve health outcomes [20]. Although most of the daily stressors within maternity work settings are unavoidable, the ability of providers to cope with and manage these stressors can directly influence their mental health status [28]. However, there is limited research on the causes of burnout and coping mechanisms adopted by maternity providers in SSA to guide the development of effective interventions to reduce or prevent burnout [13, 29]. Further, in-depth qualitative studies, which explore the reasons for, consequences of, and adaptive strategies to stress and burnout, are even more limited [30–32]. Studies that qualitatively explore the experiences of stress and burnout are critical for identifying potential intervention points, as well as identifying coping strategies that can be better supported. We sought to address this research gap and document sources of stress and coping mechanisms among providers of maternal health care in rural Kenya to inform future interventions and policy.

## Methods

### Research team

A team of researchers from Kenya and the US with quantitative and qualitative research expertise participated in data collection and analyses, with the oversight of the lead investigator (PA). Two field-based researchers (EN and BO) conducted all interviews and contributed to

analyses and interpretation. The remaining team members contributed to other research activities including analysis and interpretation of data.

Our team's approach to the data collection, analysis and interpretation recognizes our positionality as SSA researchers, including those in the diaspora, conducting research in Kenya. While a majority of our team includes researchers from Kenya, and specifically, the communities of our study, we recognize the ways in which our identities shape the data collection and the ways in which we interpret information. We view this as a strength of our approach: an approach that leverages local knowledge and expertise, prioritizes the engagement of those who collected the data in the full research cycle, and one that fosters discussion rather than seeking objectivity or conformity.

## Study setting and participants

The study took place in Migori County, in the Nyanza Province of Kenya, across selected facilities in the eight sub-counties. Migori County is a predominantly rural county bordered by Tanzania to the south. Migori's population is approximately 1.09 million, of which women of childbearing age constitute 263,602 of the population. There are an estimated 47,229 deliveries and 46,130 live births annually [33]. It is one of the 15 counties that account for over 60% of maternal deaths in the country; the estimated maternal mortality ratio is 673 deaths per 100,000 live births compared to the national average of 495 [34]. Migori has 8 sub-counties, each of which has one sub-county hospital, with the largest among them, serving as the County Referral Hospital. There over 200 government and private health facilities (inclusive of health centres and dispensaries) in the county, with about 85 conducting about 100 births or more per year [35]. The study setting has previously been described in detail [36]. Study participants included nurses, midwives, clinical officers (non-physician clinicians), medical doctors, and support staff tasked with care provision in maternity units.

## Data collection

This was a convergent mixed-methods study in which we collected quantitative data from n = 101 maternity providers and qualitative data from a subset of n = 31 providers, as previously described [37]. We purposively recruited providers from 30 health facilities with the highest recorded births in the county—inclusive of the County Referral Hospital, all the sub-county hospitals, and two to three other facilities (including government/mission/private hospitals or health centres) in each of the eight sub-counties. Surveys and interviews were conducted by two female research assistants, with bachelor's degrees, trained in qualitative research and survey data collection and had experience doing both. The research assistants visited the selected health facilities and introduced the study and obtained a list of all maternity staff as the initial list of eligible participants. Participants were identified in consultation with health administrators at the facilities, following permission from the County Health Directorate. Providers in each facility were then contacted, informed about the study objectives, invited to participate, and interviewed if willing to provide written informed consent. The goal in each facility was to recruit one or two doctors or clinical officers (if the facility had any), two or more nurses depending on the number of nurses available, and one or two support staff (nurse aids and cleaners) for the structured interviews. All providers in the maternity unit in the selected facility were eligible to participate and all present on the day of the interview were invited to participate given most facilities had fewer than the target number of providers available. All invited agreed to participate, giving a response rate of 100%.

Data were collected in person, using programmed tablets and Redcap. All approached providers and support staff agreed to partake in the survey. Quantitative data were obtained from

a cross-sectional survey with providers from across n = 30 facilities in the county. Respondents were administered a multidomain questionnaire programmed in Redcap, covering demographics, person-centred maternity care, stress, and bias, among others, during interviews which lasted 40 to 60 minutes. Several of the survey domains (not used in this manuscript) are measures previously validated in similar settings [38–40]. The questionnaire was piloted with five (n = 5) providers in the county prior to the actual survey. Surveys were conducted in English, Swahili, or Luo in private locations at the facility. We report only on the survey results related to sources of stress and coping mechanisms; other results from the survey and methodology have been published extensively [20, 37].

Qualitative data are from in-depth interviews using a semi-structured interview guide. Interviews were audio-recorded and lasted about 30–60 minutes each. Both surveys and interviews were conducted in-person, in a private space within the health facility by BO or EN, in the respondents preferred language (Luo, English, or Kiswahili).

## Data analysis

We conducted simple descriptive analysis of the survey data related to sources of stress and coping mechanisms. For qualitative data, we used a thematic analysis approach where we drew on the qualitative philosophy and values of reflexive thematic analysis integrated with codebook thematic analysis to guide our team-based approach [41]. As noted for reflexive thematic analysis, we acknowledged that meaning and knowledge are situated and contextual, and researcher subjectivity is a *resource* for knowledge production [41]. The codebook approach was, however, integrated for practical reasons to enable our inclusive team-based approach, which includes participation from team members who have varying levels of qualitative research experience [41]. We believe this team-based approach improved the quality of our process and findings.

Our analysis approach involved first developing a coding framework based on the interview guide. Transcripts were then loaded into Dedoose software, where the coding framework was applied to an initial set of transcripts by three team members (PA, EN, BO). The deductive coding framework was iteratively refined following coding of the initial set of transcripts. The team met to discuss and resolve inconsistencies. Few new inductive codes were generated following the coding of half of the transcripts, suggesting data saturation. Analytic and reflexive memos were used to develop themes. All transcripts were coded; codes were queried to generate code reports with attached texts, which were summarised in extended analytical memos where we considered both the semantic (surface) and latent (underlying) meaning of the text. These memos were then reviewed to identify themes which were organized using the socio ecological (SE) framework—which highlights multiple levels of influence—individual, interpersonal, organizational/health systems levels—on health outcomes [42, 43]. Representative quotes were selected to illustrate each theme. The analysis was an iterative process with findings discussed by the full team and the write up refined to provide an accurate and clear representation of the data. In analysing the qualitative data, we focused on salience rather than frequency [44]. The survey data is presented alongside the qualitative data to illustrate the frequency of various sources of stress and coping mechanisms. Integration is achieved in this convergent mixed-methods design, with an interview guide developed to illustrate, illucidate, and explain the survey findings; interviewees sampled from the survey participant list; and then at the interpretation and reporting level with both qualitative and quantitivate findings integrated in the results in the discussion [45, 46].

### Ethical approval

The study was reviewed and approved by the University of California, San Francisco Institutional Review Board (IRB number 17–22783), the Kenya Medical Research Institute Scientific and Ethics Review Unit (SERU 3682), and by the Kenya National Commission for Science, Technology & Innovation (NACOSTI). Approval for the study within Migori was granted by the County Commissioner and the County Director of Health. Written informed consent was obtained from all participants.

## Results

### Participant demographics

Most participants worked in public hospitals (43.6%) and health centers (43.6%), were nurses or midwives (61.4%), and had on average 6.7 years of experience. Most were female (62.4%), on average 33.7 years old, married (74.3), had attained at least a college education (82.2%), and came from Migori County (66%). About 40% had a monthly salary of between 10,000 and 50,000Kenyan Shillings (~$100 to $500) and 39% made 50,000 Kenyan Shillings or more. The demographic breakdown is similar for those who participated in only the IDs, as summarized in Table 1.

### Quantitative results

Most participants (54.9%) reported high workload and lack of supplies or equipment (14.5%) as their primary source of stress (Table 2). Secondary sources of stress included high workload (61.4%), lack of supplies and equipment (36.6%), poor salary (31.7%), attitudes of patients

**Table 1. Participant characteristics.**

| | Survey (N = 101) | | Interviews (N = 31) | |
|---|---|---|---|---|
| | No. | % | No. | % |
| Facility | | | | |
| Public Hospital | 43 | 42.6 | 18 | 58.1 |
| Public Health Centre/Dispensary | 44 | 43.6 | 9 | 29 |
| Mission/Private Hospital | 14 | 13.9 | 4 | 12.9 |
| Cadre/position | | | | |
| Nurse/Midwife | 62 | 61.4 | 18 | 58.1 |
| Clinical officer/Doctor | 16 | 15.8 | 3 | 9.7 |
| Support staff | 23 | 22.8 | 10 | 32.3 |
| Years as provider: mean (SD) | 101 | 6.74 (5.3) | 31 | 6.7(5.9) |
| Female | 63 | 62.4 | 21 | 67.7 |
| Age: mean (SD) | 101 | 33.7(6.7) | 31 | 34.4(7.0) |
| Married | 75 | 74.3 | 23 | 74.2 |
| College and above Education | 83 | 82.2 | 22 | 71 |
| From County | 67 | 66.3 | 23 | 74.2 |
| Monthly salary [a] | | | | |
| > 10,000Ksh | 20 | 20.2 | 9 | 29 |
| 10,000–50,000 Ksh | 40 | 40.4 | 11 | 35.5 |
| > 50,000 Ksh | 39 | 39.4 | 11 | 35.5 |

Notes: all totals equal to 101 except those marked [a] which have missing data with a total of 99

**Table 2. Sources of stress, experiences of humiliation, and disrespect in the last year.**

| | | Total | |
| --- | --- | --- | --- |
| | | No. | % |
| *What **causes you the most stress** at work? (Only one option)* | | | |
| High workload | | 55 | 54.5 |
| Lack of supplies or equipment | | 15 | 14.9 |
| Poor salary | | 8 | 7.9 |
| Attitude of patients/ family members | | 5 | 5.0 |
| Death of mother | | 3 | 3 |
| Frequent staff turn over | | 3 | 3 |
| Attitude of colleagues | | 2 | 2 |
| Personal/Family problems | | 2 | 2 |
| Incompetence of another provider | | 1 | 1 |
| Other | | 7 | 6.9 |
| ***What else causes** you the most stress at work? (Multiple options)* | | | |
| High workload | | 62 | 61.4 |
| Lack of supplies or equipment | | 37 | 36.6 |
| Poor salary | | 32 | 31.7 |
| Attitude of patients and family members | | 24 | 23.8 |
| Frequent staff turn over | | 18 | 17.8 |
| Attitude of colleagues | | 15 | 14.9 |
| Personal/Family problems | | 12 | 11.9 |
| Incompetence of another provider | | 11 | 10.9 |
| Attitude of superiors | | 10 | 9.9 |
| Death of mother | | 10 | 9.9 |
| Death of baby | | 6 | 5.9 |
| Lack of knowledge or skills to provide needed care | | 4 | 4 |
| Other | | 20 | 19.8 |
| *Have you ever lost a mother or baby during pregnancy or childbirth?* | | | |
| No | | 59 | 58.4 |
| Yes | | 42 | 41.6 |
| *Type of death* | | | |
| Maternal death | | 7 | 16.7 |
| Still birth | | 21 | 50 |
| Neonatal death | | 13 | 31 |
| Other death | 1 | | 2.4 |
| *Did this happen in the last year?* | | | |
| No | | 18 | 42.9 |
| Yes | | 24 | 57.1 |
| *Treated by superior in a way that was disrespectful or humiliating in last year* | | | |
| No, never | | 65 | 65.7 |
| Yes, a few times | | 25 | 25.3 |
| Yes, many times | | 7 | 7.1 |
| Yes, always | | 2 | 2 |
| *Treated by colleague in a way that was disrespectful or humiliating in last year* | | | |
| No, never | | 63 | 62.4 |
| Yes, a few times | | 32 | 31.7 |
| Yes, many times | | 5 | 5 |
| Yes, always | | 1 | 1 |

*(Continued)*

**Table 2.** (Continued)

| | Total | |
|---|---|---|
| | **No.** | **%** |
| *Treated by patient in a way that was disrespectful or humiliating in last year* | | |
| No, never | 45 | 44.6 |
| Yes, a few times | 35 | 34.7 |
| Yes, many times | 18 | 17.8 |
| Yes, always | 3 | 3 |

(20.8%), frequent staff turnover (17.8%), attitudes of colleagues (14.9%), personal/family problems (11.9%), incompetence of other providers (10.9%), attitude of superiors (9.9%), and death of mother (9.9%). Nearly half of providers (41.6%) reported they had ever lost a baby or mother during pregnancy or childbirth, with 57% of the deaths in the last year. Within the last year 34.4%, 37.7%, and 55.5% of providers reported they had been treated by a in a way that was disrespectful or humiliating by their superiors, colleagues, and patients, respectively. These sources of stress are further illustrated by the qualitative data.

## Qualitative results

We used the socioecological (SE) framework [43] to present experiences of, responses to, and coping mechanisms related to stress among maternal health providers. Our findings highlight the multiple levels of influence from policy (health system levels), institutional (facility level), interpersonal (among providers and patients), to intrapersonal levels (at the individual provider level). We present six themes related to sources of stress, which are discussed in relation to their impact at one or more levels of the SE framework. Many of the stressors discussed were at the policy, health systems and interpersonal levels, while fewer stressors were identified at the intrapersonal levels.

**Policy and health-systems stressors.** We present four themes—high workload, lack of work resources, avoidable deaths, and inadequate rewards—at this level.

*High workload.* As in the quantitative data, providers prioritized discussion of stress related to high workload. This was reportedly due to inadequate staffing, high patient volume, high staff turnover, and additional job functions. Nurses and clinical officers described often being the only providers during a shift, thus having to multitask across units and cover both outpatient and inpatient services. This was reported to be stressful and limited their ability to provide high quality respectful care.

*"You find in a shift, you can be one nurse, you need to attend to both [. . .] mothers in labor, then there are those mothers who are post-natal who need to be discharged [. . .] having all these departments, and having one person manning, at times you can have some burnout [. . .] you end up not giving somebody that respect [. . .] you don't give that respectful care."* **Male, Clinical Provider, ID 1151**

Other providers reported a lack of breaks or time for themselves. As a support staff notes:

*"I don't even get time to be free when I get here in the morning [until I leave] in the evening, I don't even get time to be free."* **Female, Support Staff, ID 1232**

Additionally, providers shared how high turnover, often due to the "*rationalization process*," a national level strategy where providers are periodically reposted to other facilities, resulted in inadequate staffing, and increased stress.

> "*They have taken some staff from [X facility], so the staffing is very low, and the work is high. So sometimes you find that you get overwhelmed; you feel tired, you are stressed and whenever you are in that condition, then you will not deliver [care] well.*" **Female, Clinical Provider, ID 1081**

Support staff also echoed concerns about understaffing, relaying that at times, they are asked to support clinical care, despite lacking the training or qualifications. One nurse's aide described a stressful incident where she was unable to support a mother due to her lack of training to manage the condition.

> "*I felt so bad and so hurt because I had a feeling that this lady was dying and am [looking at] her. She could look at me and tell me 'Please just help', but I had nothing I could do. Even as she was being taken away, I could still ask myself, like, 'why did she get PPH [Postpartum haemorrhage]?' [. . .] that could be my worst day. . .*" **Female, Support Staff, ID 2211**

Some providers shared that a significant proportion of their time was often spent on non-clinical activities, particularly documentation and administrative work related to performance metrics. These administrative tasks were perceived to take away from time with patients. The combination of staff shortage and increased administrative responsibilities led to high stress as discussed by a nurse:

> "*Generally, this place there is shortage of staff and there is a lot of work, so you find yourself stressed, you want to do everything and with a lot of documentation because what you don't document is not counted as done.*" **Female, Clinical Provider, ID 2010**

The experience of being overworked and insufficient staffing, led to providers requesting, and at times bribing, to avoid being assigned to the maternity ward. As a nurse notes:

> "*When they are doing change overs for maternity, no one wants to be taken to maternity; some even bribe the administration not to be brought to maternity. [When someone is] brought by force because no one is willing to work at the maternity ward or you find yourself [here] by circumstances if you are a weakling and you can't complain.*" **Female, Clinical Provider, ID 2011**

*Lack of work resources*. This theme covers several issues providers described as sources of stress related to inadequate resources to perform their jobs. These include lack of equipment, supplies, and drugs, and inadequate patient referral systems.

Providers attributed being stressed due to the lack of proper equipment. This resulted in situations where they had to improvise with less efficient and more time-consuming alternatives.

> "*We don't have the equipment to do the services. Let's say there is no solo shots to use for vaccine, so you are forced to use a 2cc [syringe]. . .So those solo [syringes] are not there so we are using 2cc to inject the children.*" **Male, Clinical Provider, ID 1121**

In other situations, providers reported borrowing equipment from other units, including blood pressure monitors, limiting their ability to provide frequent blood pressure monitoring in the management of pre-eclampsia:

"*There is no [blood pressure] machine at the maternity ward. Maybe we have a mother who has pre-eclampsia, and she really needs to be monitored closely. There is no BP machine [so] we keep borrowing from other wards. Sometimes we go to the other ward, and they are also busy with the BP machine [. . .] we also need to know the pressure to see whether the medication is [addressing] the blood pressure.*"

Female, Clinical Provider, ID 2010

Providers noted that inadequate pharmaceutical and non-pharmaceutical supply chain resulted and shortage in supplies and medicines, ranging from basic reagents to lifesaving oxygen. Such shortages were stressful because it impacted the quality of care provided to patients and sometimes resulted in poor outcomes, even death. A provider notes:

"*Just like I said before, is lack of sufficient commodities and supplies. So there comes a patient who is so sick and is need of something, but you are so helpless, you don't have the thing. So, at times, we may end up losing the patient; but it is not your fault, but lack of supplies.*"

Female, Clinical Provider, ID 1142

The lack of supplies at the facilities also often required that women bring basic materials such as cotton wool, gloves, and warm clothes when being admitted to the hospital to deliver their babies. However, nurses reported that some mothers arrive at the facility not having these items, which compromises the quality of care. Further, these instances were perceived to impact patient-provider interactions, where providers are at times unaware of patient-level challenges, and patients expect basic supplies to be available at a public facility. Providers reported some patients were even abusive towards them, causing them stress.

"*Patients have got their expectation; this is a government hospital. You see if there are no drugs, the patient comes and tells you, and you trying to write to the patient the drug, but at the back of your mind you know very well the drug is not within the hospital. Already, psychologically, you are affected.*" **Male, Clinical Provider, ID 2102**

The lack of supplies reportedly resulted in an unofficial marketplace where some providers were reported to acquire and sell supplies to patients, often at a higher cost. Supplies such as cotton wool, which providers felt should be freely available in the facility, were reportedly sold to patients. This was especially stressful for support staff who also served as community health volunteers, as it conflicted with what they told women in the communities in their efforts to convince women to give birth at the health facility.

"*There are some nurses who have their own supplies or those that they have taken from the facility. When I see the nurse sell the mother something like a cotton wool that costs two hundred (ksh), they sell to the mother at five hundred (ksh), it really pains me [. . .] I feel like they are oppressing the poor mothers. I feel bad [. . .]. In the community, [I] teach them that these things are free, but when they come to the facility they are asked for more money for an item.*" **Male, Support staff ID 1111**

Another major source of stress was lack of transportation and poor referral systems. Providers noted the lack of ambulances within the health facilities, and the need to wait with patients and often for long periods, after making a call for an ambulance for transfers. Others shared needing to find alternate modes of transportation. This was stressful because it often delayed care and led to poor outcomes.

*Avoidable deaths.* A major source of stress identified was the constant fear of losing a patient from complications that could be managed. The scarcity of ambulances was sometimes compounded by the lack of equipment, supplies, and medicines within the ambulance, which in some instances reportedly led to death.

> *" We tried to resuscitate, and we didn't have an ambulance on standby and we called one from X referral. By the time the ambulance got here, it came without equipment, without the oxygen; the child went and when we followed up, we were told that the baby died and that it was a first-time mother [. . .] It was horrible because we really struggled and we all didn't sleep; our only hope was that if an ambulance could come, and we were very specific that we were not only asking for an ambulance but we wanted one with an oxygen where we could continue resuscitating the baby."* **Female, Clinical Provider, ID 2171**

The perception that adverse patient outcomes could have been avoided with an adequate referral system was distressing.

> *"You refer the mother, and she ends up losing the baby on the way. When she tells you that, you feel unhappy because, if we had [an ambulance] here, you would have saved the baby. But due to the cold on the way and lack of transport, taking long to wait for the ambulance to come [. . .] because when you are doing referrals [. . .] by the time she reaches Migori she would have lost the baby, so you feel unhappy."* **Female, Clinical Provider, ID 2141**

Other sources of stress included poor infrastructure and utilities, such as lack of consistent water and sanitation facilities, and power outages, which impacted the care environment, provider and patient experience, and patient outcomes.

> *"Sometimes, the electricity goes off when a mother has delivered a preterm and there is no source of heat. So, the baby ends up dying- a death that could be avoided."* **Female, Support Staff, ID 1102**

The fear of avoidable deaths made diagnoses of any complications stressful, as providers recounted traumatic experiences related to adverse events experienced by mothers and their babies. Challenges related to post-partum haemorrhaging (PPH), inability to resuscitate, and death of mothers and babies were scenarios that were reported to cause stress. A nurse discussed PPH as something that causes panic during management, and leads to stress:

> *"What stresses me particularly at this age and at this time [. . .] stress at work. Ok stress can come, for example, you conducted a delivery you have seen that you have conducted a delivery you see that mother has developed PPH, you will kind of have some serious concern. Can this mother make it? and if at all you have delivered a baby and has come out flat, you can get worried [. . .] you feel what went wrong and you can feel what happened. You can feel because the relatives, maybe they can say 'this doctor has made us to lose the child'. . ."* **Male, Clinical Provider, ID 1151**

*Inadequate rewards for efforts*. Issues under this theme include delayed salaries or inadequate salaries, poor job security, and poor promotion prospects. These issues often made providers feel their efforts were not appreciated. Provider reports of delayed salaries or inadequate salaries was especially salient for support staff who described not being paid enough, and often not paid on time, given all their work in the facilities.

*"What stresses me is the county, we work so hard, but the pay is so little. Like, we are only paid two thousand shillings per month. So, at times, I feel like quitting the job but know that I enjoy doing the work [. . .] so that is what stresses me. Because if they could have increased [my salary] to even five thousand shillings, then it would be a little [better]."* **Male, Support Staff, ID 1111**

Inadequate and delayed salaries was also stressful for providers as it impacted their ability to support their families. Further, it decreased providers' motivation for their work as they did not feel they were receiving adequate rewards or incentives.

*"The reward is so low compared to the amount of work that I am doing. I do a lot and I think everybody knows that us nurses, we do a lot that the reward that we get from it is not tallying [. . .] You know, there is an extent when you give your responsibility your all, all the energy, all the concentration, all your mind, then you find [. . .] that maybe your salary has delayed or the salary has not been paid or there are things that you are supposed to use that money to support [families] and you are not able to."* **Male, Clinical Provider, ID 1221**

Job security was noted as a stressor by contract staff who had to renew their contracts yearly. Further some providers discussed poor promotion prospects as a source of stress.

**"***. . . let's say you have work in one job group for longer period and you need to be promoted and you cannot be promoted, you become demoralized"* **Female, Clinical Provider, ID 1041**

**Interpersonal.**   Themes related to providers' interactions with other providers and patients are captured at the interpersonal level. These include perceived negative attitudes of superiors and colleagues as well as patients, and patients delayed care seeking.

*Perceived negative attitudes of supervisors and colleagues*. Some providers discussed stressful situations related to conflict with leadership and resulting supervisory visits described as fault-finding missions. Others complained about an unsupportive leadership style which subjected providers to increased stress. A provider recalled:

*"I got into a conflict with a hospital [leader] recently. My son was sick, and I was on duty and for you to be given a sick leave, it is a process and at that time I was busy. I tried to get someone to work for me together with another qualified nurse who was on duty. This ended up bringing some trouble for me. [Leadership] was trying to say that I neglected my duty. . .and even reported me to the county [. . .] I even felt like I should leave this job and go to rest."* **Female, Clinical Provider, ID 2011**

Providers also discussed negative feedback from supervisors that was perceived to fuel stress. Superiors were perceived to focus on the negatives, neglecting the positives, which was described as stressful.

*"There was this particular day when I felt so bad [. . .] my superior said that I usually come to work late, and I close work early; [. . .] In fact, a day before, [my superior] came around 4:52pm to check whether I was around [. . .], I found [her] standing at a strategic point maybe waiting for me to see whether I was really in, or [if I was not at the duty post and was conspiring with my colleague]. I felt bad that I almost cried. It really hurt me."* **Female, Clinical Provider, ID 1181**

Similarly, other providers noted stress due to the negative attitudes and comments from other colleagues that was discouraging. One provider recalled:

*". . . I like spending a lot of time in the hospital and you end up seeing clients past 8:00pm then the following time you come late, you try to explain to someone that 'I really left late yesterday', they never take it positive. So sometimes you hear some negative comments from colleagues that you are always coming late. So, you see, some of those things they discourage you."* **Female, Clinical Provider, ID 2171**

*Perceived negative patient attitude and behaviour.* Providers also described the attitude and behaviour of patients as causing stress because it impacts care provision and outcomes. Providers were especially frustrated by patients who did not follow instructions such as pushing the baby when they were instructed not to push or refusing referrals.

*". . .this mother was very uncooperative, and she started pushing at 6 cm dilation. So, I took this mother and explained to her the dangers of pushing earlier and she was not listening, so she kept on pushing and as I was doing my vaginal exam, it was still 6 cm [. . .] the cervix was tight and thick. At the same time, on my examining finger, I see meconium stain grade three-sign of foetal distress."* **Female, Clinical Provider, ID 1081**

*Women's delayed care seeking behaviour.* Women first attempting delivery at home or with Traditional Birth Attendants (TBAs) before coming to the facility was described as stressful because it complicated the management of otherwise simple cases. For example, the traditional medicines provided by TBAs were perceived to act similar to oxytocin, putting the mother at risk of an overdose when labour was later augmented at the facility. Further, women visiting TBAs or attempting home births prior to presenting at the facility for emergency care, limited providers' ability to provide timely and efficacious care. As one clinician notes:

*"This area is chaotic for TBAs- they start these herbs that in their opinion hastens the labor and they only refer to the facility when they think that there is a problem they can't handle [. . .] if the client comes and you can clearly see the herbs, like they have them in the mouth and some are entered in the perineum -so you can see them after examination [. . .] they are meant to work like oxytocin, you also want to put in an oxytocin which is an intervention for you; so it's like you are doubling the dose and you are risking tears or even raptures to the uterus."* **Female, Clinical Provider, ID 1171**

**Individual level.**   Fewer individual-level stressors were discussed, with most in response or in relation to stressors at the health systems and at the intrapersonal levels. Thus, we present below, non-work-related stress and other expressions of stress that intersect with the individual experience.

*Non-work-related stress.* A few providers discussed challenges at home and with their families that often impacted their attitudes, perceptions, and functioning at the workplace. These challenges included health or death of family members, marital and financial conflicts, and problems with their children. One support staff discussed how home situation sometimes influences her mood at work:

*". . . I love children but apart from my work, I don't see like my life is moving ok [. . .] I separated with my husband, [and] what makes me sad sometimes is you could be talking to children [at work] and you are happy, but I ask myself now how does my own children survive? Even when I was happy, I just feel sad."* **Female, Support Staff, IDI 2072**

Another support staff discussed how conflict with her husband led to stress:

*"I mostly get stressed only when am at home because of the shouting's and fighting from my husband. Like now, he married a second wife and am the first wife. So, the husband keeps quarrelling with me, this second wife is even older than me in age and I can't say anything about it. If I do, the man [husband] wants to beat me and this makes me so stressed."* **Female, Support Staff, ID 2061**

Further, sometimes stress at home may providers to mistreat patients, especially those they perceive to be challenging.

*"If you have stress from home, then you will not treat the patients the same; maybe some you treat well, but you just feel like this other patient is annoying, so you just end up treating her in a cruel way. . ."* **Female, Clinical Provider, ID 1181**

*Effect of stressors on individual providers.* Providers described how various stressors affected them individually. These included feelings of frustration, hurt, sadness, demotivation, exhaustion, demoralization, discomfort, incompetence, loss of control, and other negative thoughts or actions.

*"I felt so sad, I lost vision until I felt like I was a mad man walking down the streets"* **Female, Support Staff, ID 1102**

*". . .for me, if I learn that the patient is not comfortable with me, I will feel bad and ask myself if I have done something wrong [. . .] that definitely makes me feel uncomfortable and disturbed because now you are not able to do your job holistically."* **Female, Clinical Provider, ID 2011**

Lack of supplies, drugs, and equipment and poor referral system described above often frustrated the providers, especially when this happened for prolonged periods and interfered with patient care. Losing a baby evoked feeling of sadness, especially if the baby could have been saved with the right equipment or supplies.

*"When you are doing referrals, and this place [referral facility] is distant, by the time [the mother] reaches [there], she would have lost the baby, so you feel unhappy."* **Female, Clinical Provider, ID 1142**

Even when they managed to save the baby and were happy with the outcome, the stress of the situation itself, often left providers feeling exhausted.

*"I also felt challenged because in the process I realized the things that I didn't do promptly, because you know saving the baby at that time really needs speed and equipment; there are things that maybe you have to improvise, like, I can remember the time where the baby had already picked and we needed to put her on oxygen but what we realized we didn't have the nose pumps to deliver throat oxygen directly, so we had to use some improvised method, which was not very efficient [. . .] such kind of an experience leaves you very weary and tired."* **Female, Clinical Provider, ID 1222**

Although most providers discussed the effect of these stressors in relation to patient outcomes, some discussed potential effects of the stressors on providers behaviour such as intention to quit their jobs.

*"I feel very bad; I end up thinking of so many things like even quitting this job but again I look at the future. It is only this job that stresses me."* **Female, Clinical Provider, ID 1181**

A provider discussed the physical effects of stress on her with symptoms such as headaches and nosebleeds:

*"When I am stressed first, I get nosebleeds and I have a lot of headaches."* **Female, Support Staff, ID 2021**

Other potential outcomes of the stressors discussed included migration out of the country, marital problems, and alcoholism.

*"Most of the time the employers have not taken time to try to find out why we are flying out to work in Europe, why are most of the nurses not married and why are we drinking alcohol too much*?" **Male, Clinical Provider, ID 1221**

**Stress coping mechanisms.**   When asked how they dealt with stress, providers shared various ways they coped with their stress including talking with friends or relatives, spending time with family and friends, as well as praying, singing, and participating in religious activities. Other coping mechanisms focused on individual activities such as sleeping, listening to music, or watching television, or willing oneself to forget about the problem or stress. Table 3 below details quantitative data on activities providers used to help manage or cope with stress, with representative quotes. These data demonstrate varying loci of control, with some providers expressing an external stress management at the interpersonal level, while others utilized more individual level stress coping mechanisms.

Not acknowledging their situation as stressful also seemed to be a coping mechanism for some who reported stress was not a problem for them. One provider proudly responded that nothing stresses him, asking the interviewer to ascertain whether he 'looked' like someone experiencing stress. Similarly, others reported not giving their colleagues and the health system the opportunity to stress them by either ignoring them or coming together to find solutions to address their problems.

*"Stress? I'm not seeing anything that can give me stress; colleagues can't give me stress, MOH can't give me stress [. . .] I don't give them that chance to stress [me], when they want to stress me up, I ignore them."* **Male, Clinical Provider**, **ID *1121***

**Table 3. Coping mechanisms with illustrative quotes.**

| What do you do when you are feeling stressed or burnt out? | N | % | Illustrative quotes |
|---|---|---|---|
| Talk to friends or relatives | 37 | 36.6 | *"I can say she [a friend] is one of the persons that I will always have to [talk to]. We always spend time, we always chat, we can even chat up to midnight. We laugh a lot, yeah, it really gets my stress out."* **Male, Clinical Provider, ID 2011** |
| Pray or engage in other religious practice | 22 | 21.8 | *"I go to church and that is what pushed me to listen to gospel music. When I was in deep stress and there is this female gospel singer that really inspired me. In a nutshell, I could go to church, and from there, I could get relieved. I usually feel like I have debugged from all my heavy burdens whenever I go to church."* **Male, Clinical Provider, ID 1221** |
| Sleep/nap | 22 | 21.8 | *"When I am tired, most of the time I like sleeping on these beds [at work], I sleep then after ten minutes I wake up and then I will be okay."* **Female, Support Staff, ID 1191** |
| Spend time with family or friends | 20 | 19.8 | *"I have a husband who is so caring; the moment he realizes that am withdrawn; he is there for me because he can just tell when am stressed out. So, he tells me so many stories and the stress just go away."* **Female, Clinical Provider, ID 1181** |
| Meditate/ Take deep breaths | 9 | 8.9 | *n/a* |
| Exercise or play a game | 4 | 4 | *"Maybe like weekends, I decide to go out and play football in the evening with my friends."* **Male, Support Staff, ID 1111** |
| Drink alcohol | 4 | 4 | *"If mostly the next day am on off duty, then I can drink a couple of bottles of beer then I feel good though it's a bad coping mechanism, but it forces me to."* **Male, Clinical Provider, IDI 2141** |
| Others: <br> • Talk to colleagues <br> • Rest/relax <br> • Watch television or movies <br> • Eat <br> • Drink water <br> • Use of phone <br> • Isolate oneself <br> • Do something else to distract <br> • Smoke | 46 | 45.5 | *"…when I am at work, I feel good because I share with my colleagues, and I forget about my problems…"* **Female, Support Staff, ID 2211** <br><br> *"When I know I am frustrated I don't even attend to patient. I relax. I can even ask for time off and stay at home."* **Female, Clinical Provider, ID 1112** <br><br> *"Just sitting watching. Like I have said am a fan of watching movies, as long as I have sat and staring at the television."* **Female, Clinical Provider, ID 2172** |

Notes: Percentages may not add up to 100 since providers mentioned multiple stress reduction and coping strategies. N/A means the strategy did not come up in the qualitative interviews.

*"[Unless] I carried the stress from home to work, but I don't see anything at work that causes me stress. I would have stress if I were not in good terms with the staffs, but in case there is a problem let's say shortage of something we come together as a team and solve it, and therefore that's why I say at work, I don't have any stress."* **Female, Clinical Provider**, **ID *2141***

## Discussion

We found various contributors to stress at the health systems, intrapersonal, and individual levels, with many of the stressors identified at the health systems and policy level. Health system and policy level factors led to high workload and lack of resources, which were the main drivers of stress—compounded by the stress of avoidable deaths. Implementation of policy strategies such as "*rationalization*", where providers were often rotated from one health facility to another, increased provider stress, as it created a skills vacuum and an environment of constant change. In instances where system-failures resulted in negative patient outcomes,

providers were especially stressed and demoralized. Other stressors were attributed to perceived inadequate rewards, including insufficient and delayed pay; negative attitudes and behaviors of superiors, colleagues, and patients; and managing maternal and fetal emergencies and complications without the adequate resources. While many of the stressors were at the policy and interpersonal levels, coping strategies were overwhelmingly at the intrapersonal and interpersonal levels, and included reaching out to others, seeking help from a higher power, and dealing with it by oneself through self-distraction, meditation, relaxation, and exercise.

High work load, unfavourable work conditions, role ambiguity, and workplace conflict, and low rewards/incentives are all previously well-documented stressors, [5, 6] further confirmed by our findings. High workload and lack of resources are unsurprisingly the main drivers of stress, as would be expected in resource-constrained environments. This is particularly salient in Kenya, where health systems are stretched and where there is approximately one hospital bed per 1,000 people and 1.5 medical doctors and 8.3 nurses/midwives per 10,000 people [47, 48]. The lack of work resources and inadequate rewards are evidenced by frequent and long standing threats of and actual labor strikes by various cadres of health care workers, including during the COVID-19 pandemic [49–51]. Resource shortages were reported to lead to an illicit marketplace where providers felt patients were being taken advantage of, and felt conflicted in their allegiance to their fellow providers and to patients, leading to added stress.

The high pressure and high stakes work environment, further compounded by the occurrence of avoidable adverse outcomes, is demoralizing to providers. The confluence of heavy workloads, insufficient staffing, and inadequate resources means that providers often had to balance professional standards with the expectations of patients and their families. These otherwise avoidable adverse outcomes create high tension environments where providers fear blame and accusation, as has been previously documented [26, 52–54]. Providers witnessing the death of a patient due to otherwise manageable conditions, witnessing a child dying of breathing difficulties due to lack of oxygen and long-distances to well-equipped facilities, coupled with poor staff management, low remunerations and long hours have all been previously reported stressors in SSA [19, 55, 56]. Similarly, about one of four providers in our quantitative cohort reported losing a mother or a baby in the past year, potentially contributing to their existing fear of adverse outcomes and subsequent blame from patients and their families. This complex care provision setting fuels an environment where abuse, disrespect, and non-patient centred care are used as means of gaining patient compliance; in these instances, providers often justified verbally or physically abusing mothers for fear of losing the baby or mother [26, 53, 54]. Concurrent with these findings, our quantitative findings have demonstrated that high stress and burnout are associated with low self-reported provision person-centred maternity care [27]. In our published quantitative analysis using validated psychosocial measures of stress, nearly all the providers (96%) had moderate to high levels of stress and more than 80% had some level of burnout, with 20% having high levels of burnout indicative of burnout of clinical concern [20]. Our findings, concurrent with previous findings, also underscore the longstanding dissatisfaction among health care providers which remains unresolved [57, 58].

These stressors must be addressed to better support providers and improve health outcomes, especially in the context of COVID-19 and increasingly stretched health systems. While the health systems stressors appear insurmountable, our findings suggest some practical ways to address them. For example, while the high patient to provider ratio is a major driver of stress, providers shared a more pressing concern- the frequent transfer of staff across health facilities. This challenge can be addressed through better forecasting, planning, and through conducting a needs assessment of impacted facilities prior to initiating the transfers. Further, health management teams can consider minimum service periods for providers to ensure

there are no service gaps, unless necessary. Streamlining administrative work and documentation and undertaking efforts to improve the maternity unit to attract and motivate providers, can go a long way towards addressing high workloads. Further, health facilities would be better served by systems and channels to both identify, assess, and intervene on stress and burnout, especially when they are at risk of or have caused negative outcomes for providers and their patients. Finally, better supply chain management and logistical planning can help prevent supply shortage and avert adverse outcomes.

Our findings on interpersonal stressors from interactions with superiors and colleagues are concurrent with previous findings [7, 59]. Studies have also highlighted the role of communication from management on stress and burnout as well as overall satisfaction [58, 60]. Open communication channels from management and employees has also been shown to reduce stress, while poor interaction and communication among colleagues and management has been shown to result in increased stress [7, 59]. These stressors, arguably easier to intervene on as compared to the health systems stressors, have been the target of past interventions. An ethnographic study of neonatal nurses in Kenya identified flexibility, autonomy, and limited managerial oversight as a collective stress coping mechanism [61]. In our study, the experience of disrespect from a supervisor or colleague in about a third of our study population, underscores the urgent need for interventions to improve supportive supervision and a positive workplace culture. Similarly, stress emanating from patients' attitudes, non-compliance, and negative behaviours can be addressed through efforts to better understand patients. Some stressors also stemmed from non-work sources including family members, children, and partners, underscoring the multi-dimensional stressors that impact providers' mental health.

The unavoidable stressors in this environment require that providers focus on building their resilience and adapting positive coping strategies. Most of the coping strategies employed by providers in this setting are positive and have been targeted in prior interventions. Previous studies have also highlighted the role of problem solving, positive re-appraisal, religiosity, optimism, and the maladaptive strategy of self-distraction [17, 62–64]. Providers sought counsel with friends, family members, and others, and turned to religion or spirituality. Further, counseling centers within healthcare facilities staffed with qualified psychologists has been shown to minimize occupational stress [65, 66]. The frequent exposure of maternity providers to adverse events necessitates the availability of mental health support that is accessible and responsive. Of note, only 13% of providers in our study reported ever participating in a stress management training [27], while only a few providers reported physical exercise, meditation, taking deep breaths or other coping mechanisms. Providers need to be introduced to evidence-based coping mechanisms and dissuaded from adapting negative coping behaviors. However, individual response strategies to manage stress, are limited in the context of broader health systems challenges.

Resource-rich settings have demonstrated evidence for interventions to improve health care worker stress and burnout, including individual-focused strategies (stress management, yoga, medication), structural or systems level strategies (workflow management, scheduling), group and individual based therapy (cognitive behavioral therapy), as well as digital/mobile interventions [67–70]. However, in SSA specifically, there is a scarcity of data on interventions to address healthcare providers' stress and promote their well-being. Among the few interventions, efforts to improve working conditions such as better remuneration, increasing essential materials, and improved communication with management have been shown to be effective [19, 55, 56, 71]. Our findings, combined with the increasing disease burden, impact of COVID-19, and limited resources in SSA, all underscore the need for interventions to prevent and address stress and burnout, while nurturing resilience and improving governance to empower health care systems [13, 72]. Similar findings from Kenya, in addition to resource

allocation, highlight the role of conflict resolution and incentives to assist health workers in coping with work-related stress [19]. Finally, the findings underscore the important role occupation health can play in addressing not just workplace safety and health, in a way that encompasses the mental health of providers.

## Strengths and limitations

The strength of the study is that the analysis and interpretation involved the full study team including those who collected the data, strengthening the rigor of the study and validity of data interpretation. Further, our findings present both quantitative and qualitative data, to strengthen findings and provide further elucidations for quantitative observations. However, data were collected in only one predominantly rural county in Kenya, thus limiting the generalizability. Findings from this setting may not be applicable to other settings. Nevertheless, data were gathered across many facilities representing government, private and other health centres, and among varied informants including a range of cadre responsible for maternal health care provision and supportive roles. This increases the generalizability of the findings. Additionally, because the semi structured IDI guides focused on sources of stress within the context of care provision, narratives overwhelmingly are about health systems level challenges. Thus, limited data on individual-level stressors among providers may be due to the way the guide was structured; although this does not reduce the value of the findings with regards to work related sources of stress.

## Conclusions

The perennial stressors identified within health care systems calls for the integration of programs to reduce the sources of stress for healthcare workers and promote positive coping mechanisms. Findings underscore the need to address various issues at the health systems, organizational, interpersonal, and individual levels. Strategies are needed to support staff retention, provide adequate resources to deliver care, provide incentives to motivate providers, and promote supportive supervision and a respectful workplace culture. Further, mental health support to enhance personal coping strategies, including an appraisal of effective interventions, are urgently needed. To improve patient outcomes, we must care for the caregivers.

## Acknowledgments

We would like to thank the research study participants and providers for sharing their insights and time. Further, we wish to thank the administrators and county leadership for providing their support for this study.

## Author Contributions

**Conceptualization:** Patience A. Afulani.

**Data curation:** Monica Getahun, Edwina N. Oboke, Beryl A. Ogolla, Joyceline Kinyua, Linnet Ongeri, Mona Sterling, Patience A. Afulani.

**Formal analysis:** Monica Getahun, Edwina N. Oboke, Beryl A. Ogolla, Joyceline Kinyua, Linnet Ongeri, Mona Sterling, Patience A. Afulani.

**Funding acquisition:** Patience A. Afulani.

**Investigation:** Linnet Ongeri, Patience A. Afulani.

**Methodology:** Patience A. Afulani.

**Project administration:** Monica Getahun, Iscar Oluoch.

**Resources:** Iscar Oluoch, Patience A. Afulani.

**Supervision:** Linnet Ongeri, Patience A. Afulani.

**Validation:** Monica Getahun, Patience A. Afulani.

**Writing – original draft:** Monica Getahun, Edwina N. Oboke, Patience A. Afulani.

**Writing – review & editing:** Monica Getahun, Edwina N. Oboke, Beryl A. Ogolla, Joyceline Kinyua, Linnet Ongeri, Mona Sterling, Iscar Oluoch, Audrey Lyndon, Patience A. Afulani.

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
