## [Decision Letter · Decision Letter 0]

17 Oct 2022

PGPH-D-22-01431

Sources of Stress and Coping Mechanisms: Experiences of Maternal Health Care Providers in Western Kenya

Dear Dr. Getahun,

Thank you for submitting your manuscript to PLOS Global Public Health. After careful consideration, we feel that it has merit but does not fully meet PLOS Global Public Health’s publication criteria as it currently stands. Therefore, we invite you to submit a revised version of the manuscript that addresses the points raised during the review process.

It is particularly important to respond to comments by reviewer #1 on the response rate for the quanitative part of the study, as well as my own comments below regarding the qualitative part (in particular addressing reflexivity, the underlying theoretical orientation and referencing your methodological choices).

We look forward to receiving your revised manuscript.

Kind regards,

M. Dylan Bould

Academic Editor

Journal Requirements:

1. Please send a completed 'Competing Interests' statement, including any COIs declared by your co-authors. If you have no competing interests to declare, please state "The authors have declared that no competing interests exist". Otherwise please declare all competing interests beginning with the statement "I have read the journal's policy and the authors of this manuscript have the following competing interests:"

Additional Editor Comments (if provided):

This is an important research question and important data - thank you for submitting to PLOS Global Public Health.

Please make sure that the reporting of qualitative data conforms to established reporting guidelines, such as

https://academic.oup.com/intqhc/article/19/6/349/1791966 or

https://journals.lww.com/academicmedicine/fulltext/2014/09000/Standards_for_Reporting_Qualitative_Research__A.21.aspx

In particular, please include a discussion of reflexivity in your manuscript.

Also consider reporting guidelines for mixed methods work such as Mixed Methods Article Reporting Standards (MMARS).

Your approach to considering both the semantic and latent meaning of the text gathered for your qualitative analysis seems appropriate for this research questions. I would however like to see more detail on your analytic approach - can you please reference the methods to other literature? Can you please expand on what you mean by the socio-ecological framework (with reference please)? As per COREQ guidelines (#9) please specify the methodological orientation used to underpin the study.

Please detail exactly what you mean by a “clinical officer” in your context.

Reviewers' comments:

Reviewer's Responses to Questions

**Comments to the Author**

1. Does this manuscript meet PLOS Global Public Health’s publication criteria? Is the manuscript technically sound, and do the data support the conclusions? The manuscript must describe methodologically and ethically rigorous research with conclusions that are appropriately drawn based on the data presented.

Reviewer #1: Partly

Reviewer #2: Yes

Reviewer #3: Partly

2. Has the statistical analysis been performed appropriately and rigorously?

Reviewer #1: Yes

Reviewer #2: Yes

Reviewer #3: Yes

3. Have the authors made all data underlying the findings in their manuscript fully available (please refer to the Data Availability Statement at the start of the manuscript PDF file)?

Reviewer #1: Yes

Reviewer #2: Yes

Reviewer #3: Yes

4. Is the manuscript presented in an intelligible fashion and written in standard English?

Reviewer #1: Yes

Reviewer #2: Yes

Reviewer #3: Yes

5. Review Comments to the Author

Reviewer #1: Dear Editor Prof. Dylan Bould,

Thanks for the opportunity to review this manuscript.

I congratulate the authors for working on this important topic about the sources of

stress of healthcare providers and their coping mechanism especially in low resources

settings.

Based my review, I think that major revisions are needed. I have provided suggestions of changes below which I think may improve the quality of the manuscript.

Kindly,

Eugene Tuyishime, MBBS, MMed, MSc

Suggested changes

In the introduction section (line 72), I suggest using a consistent term as in the title

“maternal health providers” instead of “maternal health staff”.

In the settings section, there is a need of more details about the number of health facilities and population of the study.

In the article that the authors cited, there is information about the number of nurses, clinical officers, and doctors (respectively 32, 19, and 4 100,000 people in the county), however, readers should be able to know the expected number of each professional category (population of the study). This will allow to estimate the response rate and the distribution of responses by profession.

In addition, it is unclear if some categories of maternal health providers like anesthetists were not available in the county of study.

Furthermore, there is no information about the total number of health facilities in the county of interest which is essential in determining the generalizability of the results. The authors should provide this important information.

In the methods section, the authors should provide more details:

Line 109: which participants were from private versus public health facilities?

Line 111: How many research assistants? How were they trained?

Considering the total number of maternal health providers per each professional category (nurses, doctors, clinical officers), how many questionnaires were sent out? How many questionnaires were returned (this will allow to report the response rate in the results section below)?

Line 125: Add a sentence about how the quantitative questionnaire was validated in the current settings. Was any pilot (of the questionnaire) done? Which language was used for the quantitative survey?

In the data analysis section, a reviewer with expertise in qualitative methods may comment on the choice of the deductive method in comparison to a combined inductive and deductive method especially that there is an existing literature on this topic (I am not an expert in qualitative method, I am not the best person to review this part).

In the Ethical Approval section, the authors should add the ethical approval numbers from the 2 research ethics boards.

In the results section, the authors should provide the information on the response rate.

In the discussion section, the implication about the authors comments on the first limitation is needed after the following sentence “Nevertheless, data were gathered across many facilities representing government, private and other health centers, and among varied informants including a range of cadre responsible for maternal health care provision and supportive roles.”

The authors may consider changing the section of limitation to “strengths and limitations” and start with the following sentence about the study’s strength “ line 680: The strength of our study is that our analysis and interpretation involved the full study team including those who collected the data, strengthening the rigor of the study and validity of data interpretation.” Then continue with limitations.

The first sentence of the conclusions “The perennial stressors within health care systems that we and others have identified, calls for the integration of programs to reduce the sources of stress and promote positive coping mechanisms.” should be re-written to reflect the main findings. The information about the authors “we and others” doesn’t need to be written in the first person.

Reviewer #2: Monica Getahun and friends conducted a study about Sources of Stress and Coping Mechanisms, Experiences of Maternal Health Care providers in Western Kenya .

The findings suggested reported stress are due to high workloads , lack

of supplies ,poor salary , attitudes of colleagues and superiors , attitudes of

patients and adverse outcomes . They have discussed factors contributing to each

and how the sources of stress affect providers and patient outcomes. In addition,

Coping mechanisms adopted by providers were captured under three themes: addressing stress by oneself, reaching out to others, and seeking help from a higher power.

Their findings emphasized the need to address organizational, interpersonal, and individual level stressors.

I found the manuscript extremely relevant and well organized which made it easy to follow.

1.The manuscript could suggest incorporating occupational health’s relevance to the extent of it’s legal implication.

2. The discussion could have included experience of well developed nations in combating workplace stress and Coping Mechanisms.

3. Line 482 Non – work related stress as a cause of mistreatment of patients were mentioned by females responders only. It would be interesting to find out what male respondents response was.

4. LINE 360 The multidimensional cause of undesired effect on mother and/or newborn is extremely misunderstood that the providers fear as a wrong perception of incompetence.

5.Line 563 Policy level stressors could be well elaborated.

6. If hidden reporting mechanism of occupational stress and it’s possible causes existed as well as , reporting of colleagues with burnout to the extent of endangering patients, could be put as a suggestion.

7. Random / frequent assessment of occupational stress and burn out could be suggested.

Reviewer #3: 1.It is not clear what " maternity providers" means - it appears to imply health/clinical workers i.e doctors, nurses lab techs, pharmacists etc. Is this the case? If so, why not call them health workers? Are there other providers of care e.g CHVs, social workers, patients' relatives etc. More elaboration is need on this.

2. Are emotional and psychological as indicated in the manuscript deemed to be separate phenomena?

3. The three 'coping mechanism' themes, how were they arrived at?How are they distinct?

4. What new information does this study give that is not already more or less documented and quite well known? The authors would benefit from showing the unique information that they are bringing to this subject matter

5. Small grammatical errors that need correcting

6. PLOS authors have the option to publish the peer review history of their article (what does this mean?). If published, this will include your full peer review and any attached files.

**Do you want your identity to be public for this peer review?** For information about this choice, including consent withdrawal, please see our Privacy Policy.

Reviewer #1: **Yes: **Eugene Tuyishime, MBBS, MMed, MSc

Reviewer #2: **Yes: **Rediet Shimeles Workneh

Reviewer #3: **Yes: **Judy N Khanyola

---

## [Decision Letter · Decision Letter 1]

12 Jan 2023

Sources of Stress and Coping Mechanisms: Experiences of Maternal Health Care Providers in Western Kenya

PGPH-D-22-01431R1

Dear MS Getahun,

We are pleased to inform you that your manuscript 'Sources of Stress and Coping Mechanisms: Experiences of Maternal Health Care Providers in Western Kenya' has been provisionally accepted for publication in PLOS Global Public Health.

Best regards,

M. Dylan Bould

Academic Editor

Reviewer Comments (if any, and for reference):

Reviewer's Responses to Questions

**Comments to the Author**

1. If the authors have adequately addressed your comments raised in a previous round of review and you feel that this manuscript is now acceptable for publication, you may indicate that here to bypass the “Comments to the Author” section, enter your conflict of interest statement in the “Confidential to Editor” section, and submit your "Accept" recommendation.

Reviewer #1: All comments have been addressed

2. Does this manuscript meet PLOS Global Public Health’s publication criteria? Is the manuscript technically sound, and do the data support the conclusions? The manuscript must describe methodologically and ethically rigorous research with conclusions that are appropriately drawn based on the data presented.

Reviewer #1: Yes

3. Has the statistical analysis been performed appropriately and rigorously?

Reviewer #1: Yes

4. Have the authors made all data underlying the findings in their manuscript fully available (please refer to the Data Availability Statement at the start of the manuscript PDF file)?

Reviewer #1: Yes

5. Is the manuscript presented in an intelligible fashion and written in standard English?

Reviewer #1: Yes

6. Review Comments to the Author

Reviewer #1: No new comments. My previous comments have been well addressed.

7. PLOS authors have the option to publish the peer review history of their article (what does this mean?). If published, this will include your full peer review and any attached files.

**Do you want your identity to be public for this peer review?** For information about this choice, including consent withdrawal, please see our Privacy Policy.

Reviewer #1: **Yes: **Eugene Tuyishime, MBBS, MMed, MSc
